# FlexVAR: Flexible Visual Autoregressive Modeling without Residual Prediction

**Siyu Jiao**[1] **Gengwei Zhang**[2] **Yinlong Qian**[3] **Jiancheng Huang**[3] **Yao Zhao**[1]
**Humphrey Shi**[4] **Lin Ma**[3] **Yunchao Wei**[1†] **Zequn Jie**[3†]

[1] Institute of Information Science, Beijing Jiaotong University
[2] University of Technology Sydney
[3] Meituan
[4] Georgia Institute of Technology

Code: FlexVAR

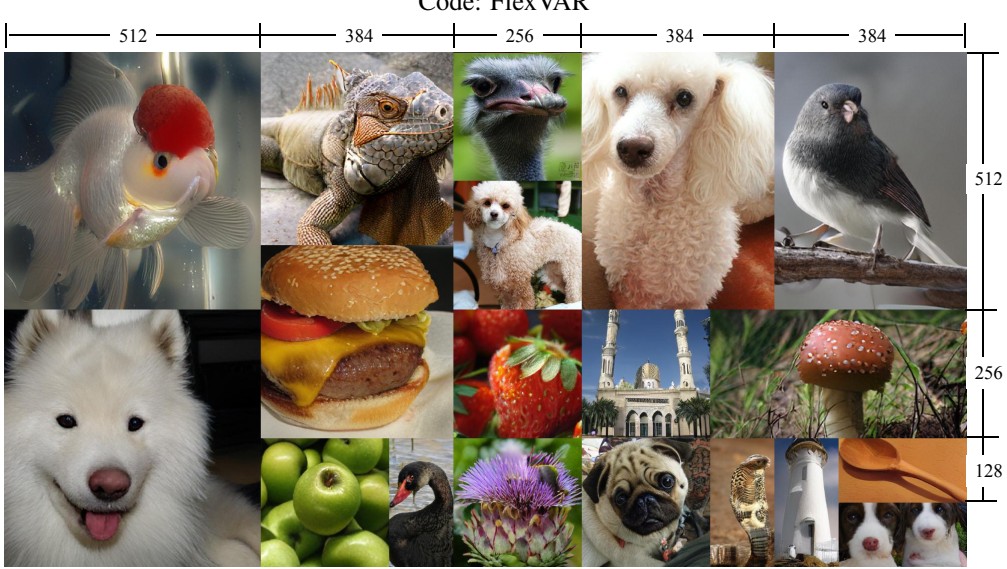

Figure 1: Generated samples from FlexVAR-$d$24 (1.0B). FlexVAR generates images with various resolutions and aspect ratios, even though it is trained with a resolution of $\leq$ 256×256.

## Abstract

This work challenges the residual prediction paradigm in visual autoregressive modeling and presents FlexVAR, a new Flexible Visual AutoRegressive image generation paradigm. FlexVAR facilitates autoregressive learning with ground-truth prediction, enabling each step to independently produce plausible images. This simple, intuitive approach swiftly learns visual distributions and makes the generation process more flexible and adaptable. **Trained solely on low-resolution images** ($\leq$ 256px), FlexVAR can: (1) Generate images of various resolutions and aspect ratios, even exceeding the resolution of the training images. (2) Support various image-to-image tasks, including image refinement, in/out-painting, and image expansion. (3) Adapt to various autoregressive steps, allowing for faster inference with fewer steps or enhancing image quality with more steps. Our 1.0B model outperforms its VAR counterpart on the ImageNet 256×256 benchmark. Moreover, when zero-shot transfer the image generation process with 13 steps, the performance further improves to 2.08 FID, outperforming state-of-the-art

---

[†]Corresponding author

39th Conference on Neural Information Processing Systems (NeurIPS 2025).

autoregressive models AiM/VAR by 0.25/0.28 FID and popular diffusion models LDM/DiT by 1.52/0.19 FID, respectively. When transferring our 1.0B model to the ImageNet 512×512 benchmark in a zero-shot manner, FlexVAR achieves competitive results compared to the VAR 2.3B model, which is a fully supervised model trained at 512×512 resolution.

# 1 Introduction

Autoregressive (AR) models aim to learn the probability distribution of the next token, offering great flexibility by generating tokens of any length. This design brings significant advancements in the field of Natural Language Processing (NLP), demonstrating satisfactory generality and transferability [4, 30, 31]. Concurrently, the computer vision field has been striving to develop large autoregressive models [28, 27, 1, 43, 29, 38]. These models employ visual tokenizers to discretize images into a series of 1D tokens [34, 22, 49, 52, 41] or 2D scales [44, 51, 42, 37, 25] and then utilize AR to model the next unit. However, these image autoregressive models typically output images at a single resolution, **the flexibility of AR has not yet been realized.**

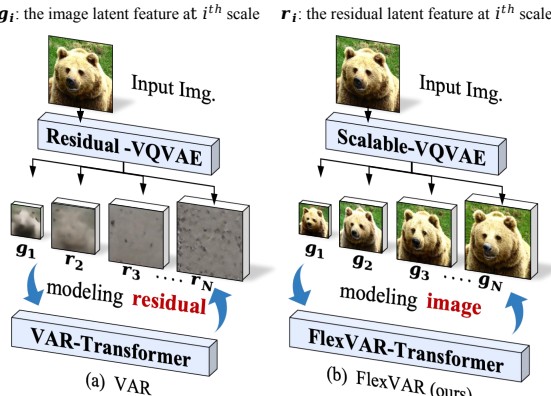

Figure 2: Comparison between VAR [44] and our FlexVAR. VAR predicts the GT[1] in step 1 and then predicts the residuals relative to the GT in all subsequent steps. Our FlexVAR predicts the GT at each step.

Recently, in image generation, VAR [44] has pioneered scale-wise autoregressive modeling, completing image autoregression based on 2D sequences. This approach predicts the next scale rather than the next token, thereby preserving the 2D structure of images and mitigating the issue of limited receptive fields in 1D causal transformers. Specifically, VAR predicts the ground-truth (GT)[1] of the smallest scale in the first step. Subsequently, at each step, it predicts the residuals of the current scale and the prior one. Finally, the outputs of each scale are upsampled to a uniform size and undergo weighted summation to generate the final output, as illustrated in Fig. 2(a). Successors [51, 42, 37, 25] have all adopted the residual design, assuming it to be effective. Although this technique achieves commendable performance, it encounters a primary challenge: The residual prediction relies on a rigid step design, restricting the flexibility to generate images with varying resolutions and aspect ratios, thus limiting the adaptability and flexibility of image generation. Meanwhile, residuals at different scales often lack semantic continuity, and this implicit prediction approach may limit the model's capacity to represent diverse image variations.

In this work, we examine the necessity of residual prediction in visual autoregressive modeling. Our intuition is that, in scale-wise autoregressive modeling, the ground-truth value of the current scale can be reliably estimated from the prior series of scales, rendering residual prediction (*i.e.*, predicting the bias between the current scale and the preceding one) unnecessary. Notably, predicting GT ensures semantic coherence between adjacent scales, making it more conducive for modeling the probability distribution of the scale. Additionally, this structure can output reasonable results at any step, breaking the rigid step design of the residual prediction and endowing autoregressive modeling with great flexibility.

Motivated by this, we systematically design the paradigm of visual autoregressive modeling without residual prediction, referred to as FlexVAR. Within FlexVAR, the ground-truth is predicted at each step instead of the residuals. Specifically, we design a scalable VQVAE tokenizer with multi-scale constraints, enhancing the VQVAE's robustness to various latent scales and thereby enabling image reconstruction at arbitrary resolutions. Then, the FlexVAR Transformer learns the probability distribution of a series of multi-scale latent features, modeling the ground-truth of the next scale, as shown in Fig. 2(b). Additionally, we propose scalable 2D positional embeddings, which incorporate

---

[1]To avoid confusion, we use ground-truth (GT) to represent image latent feature, and residual to represent residual latent feature.

2D learnable queries initialized with 2D sin-cosine weights. This approach enables the scale-wise autoregressive modeling to be extended to various resolutions/steps, including those beyond the resolutions/steps used during training, as shown in Fig. 1.

In a nutshell, this non-residual modeling approach ensures continuous semantic representation between adjacent scales. Simultaneously, it avoids the rigid step design inherent in residual prediction, significantly expanding the flexibility of image generation. FlexVAR can (1) generate images of various resolutions and aspect ratios, even exceeding the training resolutions; (2) support image-to-image tasks such as in/out-painting, image refinement, and image expansion without the need for fine-tuning; (3) enjoy flexible inference steps, allowing for accelerated inference with fewer steps or improved image quality with more steps.

## 2    Related Work

VQ-VAE [46] introduces a groundbreaking two-stage image generation paradigm: (1) encoding the image into a latent space and quantizing it to the nearest code in a fixed-size codebook; (2) modeling the discretized code using PixelCNN [45], which predicts the probability distribution of each code in raster scan order. This two-stage paradigm has laid the foundation for many subsequent works.

**Raster-scan Manner** Building on the aforementioned foundation, [12, 33] perform autoregressive learning in latent space with Transformer architecture. VQVAE-2 [35] and RQ-Transformer [22] use extra scales or stacked codes for next-image-token prediction. These works further advance the field and achieve impressive results. Recently, [41, 26] utilize a GPT-style next-token-prediction strategy to achieve high-quality image generation. [17] further improves this paradigm by introducing a mixture of autoregressive models, while [23] incorporates Mamba structure [13] to accelerate image generation. [48, 53, 14] combine diffusion processes into autoregressive modeling to address the information loss caused by quantization, which potentially degrades the quality of generated images.

**Random-scan Manner** Masked-prediction models learn to predict masked tokens in a BERT-style manner [9, 16, 2]. They introduce a bidirectional transformer that predicts masked tokens by attending to unmasked conditions, thus generating image tokens in a random-scan manner. This approach enables parallel token generation at each step, significantly improving inference efficiency. Specifically, [6, 5] apply masked-prediction models in class-to-image and text-to-image generation, respectively. MagViT series [50, 29] adapts this approach to videos by introducing a VQVAE for both images and videos. NOVA [8] first predicts temporal frames and then predicts spatial sets within each frame to achieve high-quality image/video generation.

**Scaling-scan Manner** VAR [44] establishes a new generation paradigm that redefines autoregressive learning on images from next-token-prediction to next-scale-prediction. VAR in parallel predicts image tokens at one scale, significantly reducing the number of inference steps. Following VAR, VAR-CLIP [51] achieves text-to-image generation by converting the class condition token into text tokens obtained from the CLIP. In terms of operational efficiency, [7] introduces an efficient decoding strategy, [37] incorporates linear attention mechanisms to accelerate image generation, and [25] designs a lightweight image quantizer, significantly reducing training costs. Regarding generation quality, [42, 36] optimize image details by using continuous tokenizers in combination with flow matching or diffusion model. Infinity [15] redefines the visual autoregressive model under a bitwise token prediction framework, remarkably enhancing generation capability and detail.

## 3    Methodology

### 3.1    Scale-wise Autoregression (Preliminary)

Scale-wise autoregressive models tokenize the input image into a sequence of multi-scale discrete image token maps $T = \{t_1, t_2, ..., t_n\}$, where $t_i$ is the token map with the resolution of $h_i \times w_i$ downsampling from $t_n \in \mathcal{R}^{h_n \times w_n}$. Each autoregressive step generates an entire token map, rather than a single token. Compared to next-token-prediction, which contains one token at each step, $t_i$ contains $h_i \times w_i$ tokens and is able to maintain the 2D structure.

Previous approaches [44, 51, 7, 37, 25, 42] typically follow a residual prediction paradigm. They only regress the ground-truth at the first scale ($g_1$), while at subsequent $i^{th}$ scale, the residual between

the preceding scale ($g_{i-1}$) and current scale ($g_i$) is predicted. We formulate residuals ($\{r_i\}_{i=2}^n$) as:

$$r_i = g_i - \text{Upsample}^i(g_{i-1}),\tag{1}$$

here Upsample$^i$ represents upsample $g_{i-1}$ to the $i^{th}$ scale. The autoregressive likelihood is:

$$p(g_1, r_2, \ldots, r_n) = \prod_{i=1}^n p(r_i \mid g_1, r_2, \ldots, r_{i-1})\tag{2}$$

attention mechanisms (*e.g.*, Transformer [47]) are utilized to instantiate this modeling. During the $i^{th}$ autoregressive step, all preceding residuals are merged in the autoregressive model, which then predicts the probability distribution of $r_i$. The $h_i \times w_i$ tokens in $r_i$ are generated in parallel, conditioned on all preceding units. Thus, image token maps can be redefined as: $T = \{g_1, r_2, r_3..., r_n\}$. Finally, each image token map in $T$ is upsampled to $\mathcal{R}^{h_n \times w_n}$ and summarized for image generation.

## 3.2 Overview of FlexVAR

Our FlexVAR is a flexible visual autoregressive image generation paradigm that allows autoregressive learning with ground-truth prediction rather than residual, enabling to generate reasonable images at any step independently. Within our approach: (1) A scalable VQVAE tokenizer quantizes the input images into tokens at various scales and reconstructs images, as detailed in Sec. 3.3. (2) A FlexVAR transformer is trained via scale-wise autoregressive modeling, with the removal of residuals, as detailed in Sec. 3.4.

## 3.3 Quantize & reconstruct images at various scales

Mainstream VQVAE tokenizers perform well at a single resolution. However, when scaling the latent space, they often fail to reconstruct images (as shown in Fig. 3). This observation motivates us to explore a scalable tokenizer that quantizes input images into tokens at various scales and reconstructs images. Specifically, the proposed scalable tokenizer first encodes an image into multi-scale latent space, and then uses a quantizer to convert latent space features into discrete tokens, finally a decoder is used to reconstruct the original images from the discrete tokens at each scale.

**Encoding.** Given an input image $I \in \mathcal{R}^{H \times W}$, an autoencoder $\mathcal{E}(\cdot)$ [12] is used to convert $I$ into latent space $f$:

$$f = \mathcal{E}(I), \quad f \in \mathcal{R}^{C \times h \times w}\tag{3}$$

here $h = \frac{H}{16}, w = \frac{W}{16}$. We then downsample $f$ at $K$ random scales to obtain multi-scale latent features $\mathcal{F} = \{f_1, f_2, ..., f_K\}$. $f_k$ represents represents the latent feature of the $k^{th}$ downsample from $f$. $f_K$ matches the original resolution of $f$.

**Quantizing.** The quantizer $\mathcal{Q}(\cdot)$ includes a codebook $Z \in \mathcal{R}^{V \times C}$ containing $V$ learnable vectors. The quantization process $q = \mathcal{Q}(f)$ is implemented by finding the Euclidean nearest code $q^{(k,i,j)}$ of each feature vector $f^{(k,i,j)}$ in multi-scale latent features $\mathcal{F}$:

$$q^{(k,i,j)} = \left(\text{argmin}_{v \in [V]} \|\text{Select}(Z, v) - f^{(k,i,j)}\|_2\right) \in [V]\tag{4}$$

where $\text{Select}(Z, v)$ denotes selecting the $v^{th}$ vector in codebook $Z$. Based on $\mathcal{F}$, we extract all $q^{(k,i,j)}$ and minimize the distance between $q$ and $f$ to train the quantizer $\mathcal{Q}$.

**Decoding.** The multi-scale images $\hat{\mathcal{I}} = \{\hat{I}_1, \hat{I}_2, ..., \hat{I}_K\}$ are reconstructed using the decoder $\mathcal{D}(\cdot)$ [12] given $q^{(k,i,j)}$. We follow Llamagen [41] to adopt the same loss functions ($\mathcal{L}_{vae}$) to train $\{\mathcal{E}, \mathcal{Q}, \mathcal{D}\}$ at each scale without special design. Therefore, the final loss function can be formulated:

$$\mathcal{L} = \sum_{k=1}^K \mathcal{L}_{vae}\left((I_k, \hat{I}_k), (f_k, q_k)\right)\tag{5}$$

## 3.4 Visual autoregressive modeling without residual

We reconceptualize the next-scale-prediction progress from residual prediction to GT prediction. As illustrate in Fig. 2 (b). Here, each autoregressive step predicts the GT of current scale, rather than

the residual. We start by sampling $N$ multi-scale token maps $\{g_1, g_2, \ldots, g_N\}$ from latent feature $f$, each at an increasingly higher resolution $h_n \times w_n$, culminating in $g_N$ matches the original feature map's resolution $\mathcal{R}^{C \times h \times w}$. The autoregressive likelihood is reformulated as:

$$p(g_1, g_2, \ldots, g_n) = \prod_{i=1}^{n} p(g_i \mid g_1, g_2, \ldots, g_{n-1}). \tag{6}$$

During the $i^{th}$ autoregressive step, $g_i \in \mathcal{R}^{h_i \times w_i}$ contains $h_k \times w_k$ tokens are generated in parallel, conditioned on all preceding scales $\{g_1, g_2, \ldots, g_{i-1}\}$.

**Scalable Position Embedding.** VAR utilizes fix-length Position Embedding (PE) by adding learnable queries to each step and h-w coordinates. This requires both training and inference to follow a fixed number of steps and resolutions, which limits the flexibility of the autoregressive process.

In our FlexVAR, we design a 2D scalable PE ($\mathcal{P} \in \mathcal{R}^{d \times 2h \times 2w}$) adding to the h-w coordinates. It contains $2h \times 2w$ learnable queries with $d$ channels. At the $i^{th}$ step, $\mathcal{P}$ is upsampled/downsampled to match the scale of $g_i$. To ensure stability during linear interpolation across various scales, we set $\mathcal{P}$ to 2× the size of the max latent space in traning. $\mathcal{P}$ is initialized using 2D sin-cosine PE [11] to ensure the 2D positional correlation. Additionally, we experimentally find that in our ground-truth prediction paradigm, incorporating PE for step embeddings is unnecessary (Tab. 6). Therefore, we remove the step embeddings to ensure the flexibility of steps in autoregressive modeling.

**Step sampling.** During training, we randomly sample the scale size in each step to enhance FlexVAR's capability to perceive any scale. Specifically, we set the maximum number of steps to 10, fixing the scale size of the first step to 1×1 and the last step to 16×16 (corresponding to 256×256 input images), and randomly sampling the scale sizes for the intermediate steps. Each step is dropped with a 5% probability, with a maximum of 4 steps being dropped. Thus, the number of steps during training is from 6 to 10. During inference, we use a default of 10 steps: {1, 2, 3, 4, 5, 6, 8, 10, 13, 16} (same as VAR). Our experimental results show more steps yield better performance (Fig. 6).

## 4 Experiments

### 4.1 Implementation details

**FlexVAR tokenizer.** Our scalable VQVAE tokenizer is configured with a downsampling factor of 16 and is initialized with the pre-trained weights from LlamaGen [41], the codebook size is set to 8912, and the latent space dimension is set to 32. The quantization of each scale shares the same codebook. We follow the VQVAE training recipe of LlamaGen. The training is on OpenImages [21] with a constant learning rate of $10^{-4}$, AdamW optimizer with $\beta_1 = 0.9$, $\beta_2 = 0.95$, weight decay = 0.05, a batch size of 128, and for 20 epochs. $K$ is set to 5 by default, indicating that each latent space is randomly sampled into 5 different resolutions.

**FlexVAR transformer.** We provide FlexVAR in three scales, with detailed configurations for each scale provided in Tab 1. FlexVAR is trained on the ImageNet-1K 256×256 using 80GB A100 GPUs. The training process employs the AdamW optimizer with

| Model name | Layers | Params. | Heads | Dims. | Epoch |
|------------|--------|---------|-------|-------|-------|
| FlexVAR-$d16$ | 16 | 310M | 16 | 1024 | 180 |
| FlexVAR-$d20$ | 20 | 600M | 20 | 1280 | 250 |
| FlexVAR-$d24$ | 24 | 1.0B | 24 | 1536 | 350 |

Table 1: Configuration of FlexVAR.

$\beta_1 = 0.9$, $\beta_2 = 0.95$, and a weight decay rate of 0.05. The learning rate is set to 1e-4, with the training epochs varying between 180 and 350 depending on the model scale.

### 4.2 Overall Comparison

We compare FlexVAR with existing generative methods on the ImageNet-1K benchmark, including GAN, diffusion models, random-scan, raster-scan, and scaling-scan autoregressive models. As shown in Tab. 2. To ensure a fair comparison, we only present models with a size smaller than 1B. Our FlexVAR achieves state-of-the-art performance in all generative methods, and performs remarkably well compared to the VAR counterparts. Specifically, we achieve -0.45, -0.56, and -0.12 FID improvement compared with VAR at different model sizes.

Table 2: **Generative model comparison on class-conditional ImageNet 256×256**. Metrics include Fréchet inception distance (FID), inception score (IS), precision (Pre) and recall (rec). Step: the number of model runs needed to generate an image. Time: the relative inference time of VAR-$d30$ [44]. We present models with a size ≤ 1B.

| Model | FID↓ | IS↑ | Pre↑ | Rec↑ | Param | Step | Time |
|---|---|---|---|---|---|---|---|
| *Generative Adversarial Networks (GAN)* | | | | | | | |
| BigGAN [3] | 6.95 | 224.5 | 0.89 | 0.38 | 112M | 1 | – |
| GigaGAN [20] | 3.45 | 225.5 | 0.84 | 0.61 | 569M | 1 | – |
| StyleGan-XL [40] | 2.30 | 265.1 | 0.78 | 0.53 | 166M | 1 | 0.2 |
| *Diffusion Models* | | | | | | | |
| ADM [10] | 10.94 | 101.0 | 0.69 | 0.63 | 554M | 250 | 118 |
| CDM [18] | 4.88 | 158.7 | – | – | – | 8100 | – |
| LDM-4-G [39] | 3.60 | 247.7 | – | – | 400M | 250 | – |
| DiT-L/2 [32] | 5.02 | 167.2 | 0.75 | 0.57 | 458M | 250 | 2 |
| DiT-XL/2 [32] | 2.27 | 278.2 | 0.83 | 0.57 | 675M | 250 | 2 |
| *Random-scan Manner (Mask Prediction)* | | | | | | | |
| MaskGIT [6] | 6.18 | 182.1 | 0.80 | 0.51 | 227M | 8 | 0.4 |
| RCG (cond.) [24] | 3.49 | 215.5 | – | – | 502M | 20 | 1.4 |
| *Raster-scan Manner (Token-wise Autoregressive)* | | | | | | | |
| VQGAN-re [12] | 18.65 | 80.4 | 0.78 | 0.26 | 227M | 256 | 7 |
| RQTran. [22] | 13.11 | 119.3 | – | – | 821M | 68 | - |
| LlamaGen-XL [41] | 2.62 | 244.08 | 0.80 | 0.57 | 775M | 256 | 27 |
| AiM [23] | 2.56 | 257.2 | 0.81 | 0.57 | 763M | 256 | 12 |
| *Scaling-scan Manner (Scale-wise Autoregressive)* | | | | | | | |
| VAR-$d16$ [44] | 3.55 | 280.4 | 0.84 | 0.51 | 310M | 10 | 0.2 |
| FlexVAR-$d16$ | 3.05 | 291.3 | 0.83 | 0.52 | 310M | 10 | 0.2 |
| VAR-$d20$ [44] | 2.95 | 302.6 | 0.83 | 0.56 | 600M | 10 | 0.3 |
| FlexVAR-$d20$ | 2.41 | 299.3 | 0.85 | 0.58 | 600M | 10 | 0.3 |
| VAR-$d24$ [44] | 2.33 | 312.9 | 0.82 | 0.59 | 1.0B | 10 | 0.5 |
| FlexVAR-$d24$ | 2.21 | 299.1 | 0.83 | 0.59 | 1.0B | 10 | 0.5 |
| FlexVAR-$d24$ | 2.08 | 315.7 | 0.83 | 0.59 | 1.0B | 13 | 0.6 |

## 4.3 Zero-shot Comparison

**Zero-shot inference with more steps.** We use 13 steps for image generation without training, as shown in the last row of Tab. 2. FlexVAR can flexibly adopt more steps to improve image quality. By using 13 inference steps, FlexVAR further enhances the performance to 2.08 FID and 315 IS, manifesting strong flexibility and generalization capabilities. The specific steps design is detailed in the Supplementary Material.

**Zero-shot inference on ImageNet 512×512 benchmark.** We use FlexVAR-$d24$ to generate 512×512 images and evaluate on ImageNet-512 benchmark without training, as shown in Tab. 3. Surprisingly, our FlexVAR-$d24$ exhibits competitive performance when compared to VAR, despite FlexVAR being trained only on resolutions ≤ 256×256 and having only 1.0B parameters.

| Model | Training Free | FID | IS | Params. |
|---|---|---|---|---|
| BigGAN [3] | × | 8.43 | 177.9 | 112M |
| ADM [10] | × | 23.24 | 101.0 | 554M |
| DiT-XL/2 [32] | × | 3.04 | 240.8 | 675M |
| MaskGIT [6] | × | 7.32 | 156.0 | 227M |
| VQGAN [12] | × | 26.52 | 66.8 | 1.4B |
| VAR-$d36$ [12] | × | 2.63 | 303.2 | 2.3B |
| FlexVAR-$d24$ (ours) | ✓ | 4.43 | 314.4 | 1.0B |

Table 3: Zero-shot inference on ImageNet 512×512 conditional generation. **Training Free** indicates whether the model is trained at the 512×512 resolution.

## 4.4 Ablation study

We conduct ablation studies on various design choices in FlexVAR. Due to the limited computational resources, we report the results trained with a short training scheme in Tab. 4, 5, 6, *i.e.*, 40 epochs (∼ 70K iterations).

**Component-wise ablations.** To understand the effect of each component, we start with standard VAR and progressively add each design, as shown in Tab. 4:

- **Baseline:** VAR uses a residual prediction paradigm, exhibits decent performance ([1st] result), but its flexibility in image generation does not meet expectations (as described in Sec. 1).
- **Prediction type:** It is infeasible to directly convert the prediction type to GT, as seen in the [2nd] and [3rd] results. We employ the VQVAE tokenizers from VAR and Llamagen, both of which yield inferior performance. This is not surprising, as the current tokenizers lack robustness to images with varying latent space, while we force these tokenizers to obtain multi-scale latent features during training (we provide a detailed analysis in Fig. 3).
- **Tokenizer:** Our scalable tokenizer obtains reasonable multi-scale latent features during training, resulting in an improvement of -13.87 FID (the [4th] results). However, flexible image generation is not accomplished yet.
- **Position embedding:** As shown in the last result in Tab. 4, the introduction of our scalable Position Embedding (PE) provides high flexibility for image generation, and further enhances the performance to 3.71 FID.

| Pred. type | VQVAE | PE | FID | IS |
|---|---|---|---|---|
| Residual | VAR | fixed-length | 4.00 | 226.04 |
| GT | VAR | fix-length | N/A | N/A |
| GT | Llamagen | fix-length | 17.75 | 234.12 |
| GT | ours | fix-length | 3.82 | 229.35 |
| GT | ours | scalable | 3.71 | 230.22 |

Table 4: Ablation of diverse designs. We use the *next-scale-prediction* paradigm, explore the effects of different prediction types (residual/GT), VQVAE tokenizers (Llamagen/VAR/ours), and positional embedding (fix-length/scalable). N/A denotes the model does not converge during training. We report the results with model scale *-d20* trained 40 epochs on ImageNet-1K.

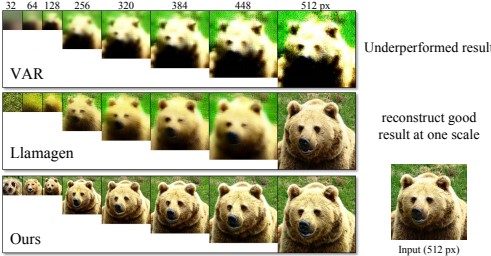

Figure 3: Compared with VQVAE [44, 41] for multi-scale reconstructing images, we downsample the latent features in VQVAE to multiple scales and use the VQVAE Decoder to reconstruct images. We upsample images < 100 pixels using bilinear interpolation for a better view.

**Reconstruct images with different VQVAEs.** In Fig. 3, we reconstruct multi-scale images by scaling the latent features in VQVAE tokenizers. Existing VQVAE tokenizers typically do not support scaling the latent features across a range of small to large scales. VAR's VQVAE [44] uses a residual-based training recipe, directly applying it to non-residual image reconstruction does not yield the anticipated results (the [1st] row). The VQVAE tokenizer from Llamagen [41] shows excellent reconstruction performance only at the original latent space, indicating that it is not feasible for scale-wise autoregressive modeling (the [2nd] row).

| Depth | Atten. type | FID | IS | Params. | Time |
|---|---|---|---|---|---|
| *-d16* | Transformer | 4.32 | 209.87 | 310M | 0.2 |
|  | Mamba | 4.22 | 200.04 | 370M | 0.2 |
| *-d20* | Transformer | 3.71 | 230.22 | 600M | 0.3 |
|  | Mamba | 3.80 | 216.45 | 700M | 0.3 |

Table 5: Ablation of the Mamba architectural. Models are trained 40 epochs ($\sim$ 70K iterations).

| Step | h-w coordinates | learnable | FID | IS |
|---|---|---|---|---|
| fix-length | fixed-length | True | 3.82 | 229.35 |
| $\times$ | fixed-length | True | 3.87 | 224.25 |
| $\times$ | scaleable | False | 3.74 | 224.04 |
| $\times$ | scaleable | True | 3.71 | 230.22 |

Table 6: Ablation of Position Embedding. $\times$ denotes that the corresponding PE is removed.

**Transfer FlexVAR to Mamba.** Recent work, AiM [23], uses the Mamba architecture for token-wise autoregressive modeling. Inspired by this, we modify FlexVAR with Mamba to evaluate the performance (Tab. 5). With similar model parameters, Mamba demonstrates competitive results compared to transformer models, indicating the GT prediction paradigm can effectively adapt to linear attention mechanisms like Mamba. However, considering that this Mamba architecture does not reflect the speed advantage, we do not integrate Mamba into our final version.

Mamba's inherent unidirectional attention mechanism prevents image tokens from achieving global attention within the same scale. To address this issue, we employ 8 scanning paths in different Mamba layers to capture global information. The specific Mamba architecture is detailed in the Supplementary Material.

**Position Embedding.** In Tab. 6, we experiment with several types of step PE and x-y coordinate PE. To make the model robust to inference steps and enable it to generate images at any resolution, we remove the fixed-length step embedding (results in the second row), and the performance showed only slight changes. We adopt a non-parametric variant, similar to ViT [11], which shows a 0.03 FID difference compared to the learnable variant.

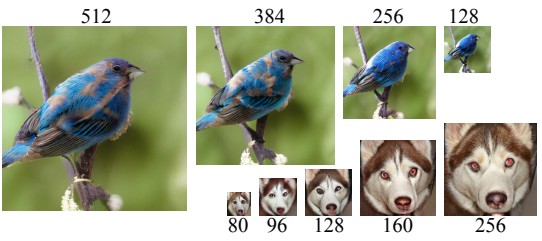

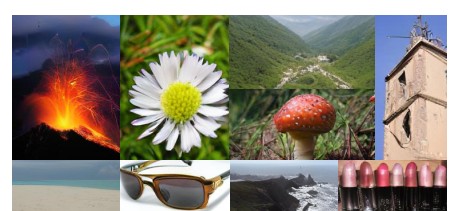

Figure 4: Generated samples from 80px to 512px. FlexVAR demonstrates strong consistency across various scales and can generate 512px images, despite the model being trained only on images ≤ 256px.

Figure 5: Generated samples with various aspect ratios. FlexVAR-$d24$ is used. FlexVAR demonstrates good visual quality across images with various aspect ratios.

**VAE Performance Comparison.** We compared the reconstruction performance of VAR and our proposed VAE in Tab. 7. The image reconstruction quality is measured by r-FID, reconstruction-FID on ImageNet validation set.

|  | Epoch | Batch-Size | rFID |
|---|---|---|---|
| VAR-VAE | 20 | 768 | 1.92 |
| FlexVAR-VAE (ours) | 10 | 128 | 3.79 |

Table 7: VAE Performance.

We train FlexVAR-VAE with fewer epochs and smaller batch sizes. We observe its reconstruction quality inferior to VAR-VAE. Thus the improvement is not due to the quality of the discrete tokens, using a more robust FlexVAR-VAE might further improve the quality of generated images

### 4.5 Analysis and Discussion

**Generate images at various resolution.** We show generated images at different resolutions using FlexVAR-$d24$ in Fig 1, 4. By controlling the inference steps, our FlexVAR can generate images at any resolution, despite being trained only on images with resolutions ≤ 256px. The generated images demonstrate strong semantic consistency across multiple scales, and the higher resolutions exhibit more detailed clarity. See the Supplementary Material for more zero-shot high-resolution generation samples and step designs.

**Generate images at various ratio.** We use FlexVAR-$d24$ to generate samples with various aspect ratios in Fig. 1 and Fig. 5. By controlling the aspect ratio at each step of the inference process, our FlexVAR allows for generating images with various aspect ratios, demonstrating the flexibility and controllability of our GT prediction paradigm. We control the height and width at each scale through approximate rounding. *e.g.*, to generate an image of size $H \times W$, the corresponding VAE latent feature size is $h \times w$, where $h = \frac{H}{16}$ and $w = \frac{W}{16}$. We adopt VAR's default set of 10 steps ($K = \{1, 2, 3, 4, 5, 6, 8, 10, 13, 16\}$) to determine the size at each scale. As a result, the $H \times W$ image corresponds to ten scales with sizes $\{\text{int}(h \times \frac{i}{16}), \text{int}(w \times \frac{i}{16})\}_{i \in K}$.

**Generate images at various step.** In Fig. 6, we investigate the FID and IS for generating $256 \times 256$ images from 6 to 16 steps with 3 different sizes (depth 16, 20, 24). As the number of steps increases, the quality of the generated images improves. The improvement is more significant in larger models (*e.g.*, FlexVAR-$d24$), as larger transformers are thought able to learn more complex and fine-grained image distributions. During training, we use up to 10 steps to avoid OOM (out-of-memory) problem. Surprisingly, in the inference stage, using 13 steps results in a performance gain of -0.13 FID. This observation indicates that our FlexVAR is flexible with respect to inference steps, allowing for fewer steps to speed up image generation or more steps to achieve higher-quality images. The details of various step designs are provided in the Supplementary Material.

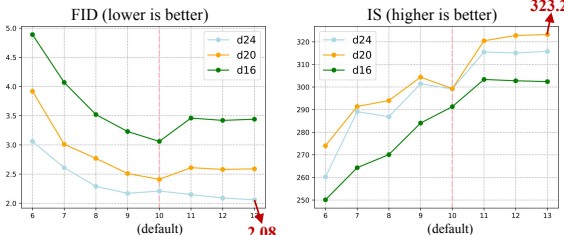

Figure 6: Zero-shot image generation at different steps (from 6 to 13 steps). FID and IS are used for evaluation. We use $\leq 10$ steps for training, and FlexVAR can zero-shot transfer to 13 steps during inference and achieve better results.

**Refine image at high resolution.** In Fig. 7, we input low-resolution images (e.g., 256px×256px) and enable FlexVAR-$d24$ to output high-resolution refined images. Despite being trained only on $\leq$ 256px images, FlexVAR effectively refines image details by increasing the input image resolution, such as the eyes of the dogs in the example. This demonstrates the high flexibility of FlexVAR in image-to-image generation.

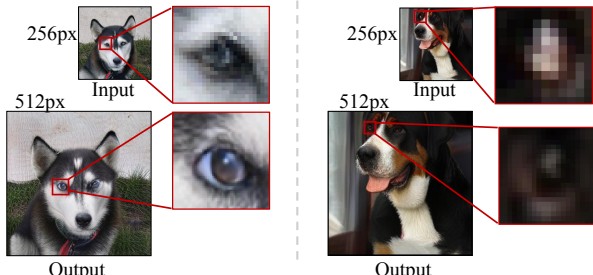

Figure 7: Zero-shot image refinement at high resolution. Zoom in for a better view.

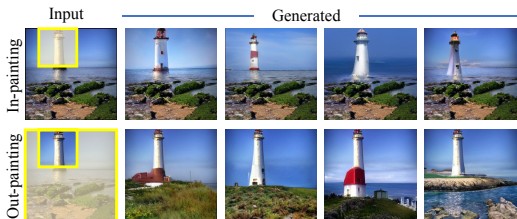

Figure 8: Zero-shot evaluation in/out-painting. The results show that FlexVAR can generalize to novel downstream tasks without special design and finetuning.

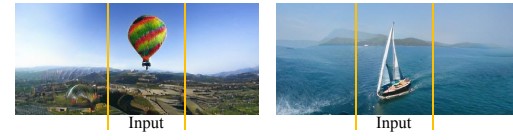

Figure 9: Zero-shot evaluation image expansion. The results show that FlexVAR can generalize to novel downstream tasks without special design and fine-tuning.

**Image in-painting and out-painting.** For in-painting and out-painting, we teacher-force ground-truth tokens outside the mask and let the model only generate tokens within the mask. Class label information is also injected. The results are visualized in Fig. 8. Without modifications to the architecture design or training, FlexVAR achieves decent results on these image-to-image tasks.

**Image extension.** For image extension, we extend images with an aspect ratio of 1:2 for the target class, with the ground-truth tokens forced to be in the center. The results are visualized in Fig. 9. FlexVAR shows decent results in image extension, indicating the strong generalization ability and flexibility of our architecture.

**Failure case.** FlexVAR fails to generate images with a resolution $3\times$ or more than the training resolution, as illustrated in Fig. 10. These cases typically feature noticeable wavy textures and blurry areas in the details. This failure is likely due to the overly homogeneous structure of the current training dataset. *i.e.*, ImageNet-1K generally lacks multi-scale objects ranging from coarse to fine, leading to errors in generating details of high-resolution objects.

We hypothesize that training the model with a more complex dataset that includes images with fine-grained details, the model might become robust for higher resolutions.

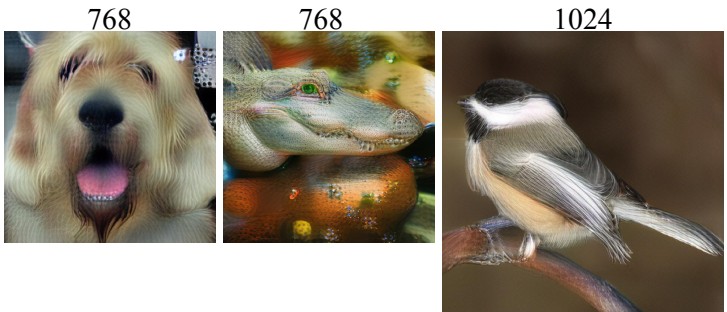

Figure 10: Failure cases at high resolutions (768 & 1024). FlexVAR shows wavy textures when generating high-resolution images. Zoom in for a better view.

## 5   Conclusion

In this paper, we introduce FlexVAR, a flexible visual autoregressive image generation paradigm that allows autoregressive learning without residual prediction. We design a scalable VQVAE tokenizer and FlexVAR-Transformer for this purpose. This ground-truth prediction paradigm endows the autoregressive model with great flexibility and controllability, enabling image generation at various resolution, aspect ratio, and inference step, beyond those used during training. Moreover, it can zero-shot transfer to various image-to-image generation tasks. We hope FlexVAR will serve as a solid baseline and help ease future research of visual autoregressive modeling and related areas.

**Limitations.** We observe that when generating images with a resolution $\geq 3\times$ larger than the training image, noticeable wavy textures appear (Fig. 10). This issue may be attributed to the homogeneous structure of the ImageNet-1K training set. We will investigate this further in future work to explore how to ensure stability in zero-shot image generation at higher resolutions.

**Broader Impact.** This research strictly follows established practices for class-to-image (c2i) model training and evaluation. Similar to most generative models, our approach may inherit biases present in the training datasets. We advocate for the responsible use of this technology and caution when deploying it in real-world scenarios.

## Acknowledgments and Disclosure of Funding

This work is supported by the National Natural Science Foundation of China (No. 92470203, U23A20314, U24B20179, 6120106009), the Beijing Natural Science Foundation (No. L242022), and the Fundamental Research Funds for the Central Universities (No. 2024XKRC082).

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

# A    Inference steps

In Tab. 8, we list the scales corresponding to different inference steps. The scales in each step are not fixed and can be flexibly adjusted during inference. Note that during training, we only limit the maximum number of steps to 10 and randomly sample the scale for each step, so the scales during the training process do not follow Tab. 8

| Reso | Step | Scale |
|---|---|---|
| 256px | 6 | {1, 2, 4, 6, 10, 16} |
| | 7 | {1, 2, 3, 5, 8, 11, 16} |
| | 8 | {1, 2, 3, 4, 6, 10, 13, 16} |
| | 9 | {1, 2, 3, 4, 5, 7, 10, 13, 16} |
| | 10 | {1, 2, 3, 4, 5, 6, 8, 10, 13, 16} |
| | 11 | {1, 2, 3, 4, 5, 6, 7, 9, 11, 13, 16} |
| | 12 | {1, 2, 3, 4, 5, 6, 7, 8, 10, 12, 14, 16} |
| | 13 | {1, 2, 3, 4, 5, 6, 7, 8, 9, 10, 12, 14, 16} |
| 384px | 11 | {1, 2, 3, 4, 5, 6, 8, 10, 13, 16, 24} |
| 512px | 12 | {1, 2, 3, 4, 5, 6, 8, 10, 13, 16, 23, 32} |
| | 15 | {1, 2, 3, 4, 5, 6, 7, 8, 9, 10, 12, 14, 16, 23, 32} |

Table 8: Scale configurations of various inference steps.

# B    Extent visual autoregressive modeling with Mamba

Unlike attention mechanisms that utilize explicit query-key-value (QKV) interactions to integrate context, Mamba faces challenges in handling bi-directional interaction. Therefore, prior Mamba-based visual autoregressive work [37] only used Mamba to model the unidirectional relationship between scales, relying on additional Transformer layers to process tokens within one scale.

In this work, we adopt a composition-recomposition strategy to obtain global information in Mamba network. Specifically, we utilize a Zigzag scanning strategy [19] over the spatial dimension. We alternate between eight distinct scanning paths across different Mamba layers (as shown in Fig. 11), which include:

- (a) top-left to the bottom-right.
- (b) top-left to the bottom-right.
- (c) bottom-left to the top-right.
- (d) bottom-left to the top-right.
- (e) bottom-right to the top-left.
- (f) bottom-right to the top-left.
- (g) top-right to the bottom-left.
- (h) top-right to the bottom-left.

# C    Qualitative results with different steps.

In Fig. 12, we show some generated samples with {6, 8, 10, 12} steps. Our FlexVAR uses up to 10 steps for autoregressive modeling during training to avoid OOM (out-of-memory), while it can naturally transfer to any number of steps during inference. The samples generated with different steps are highly similar, differing only in some details. Generally, more steps result in better image details.

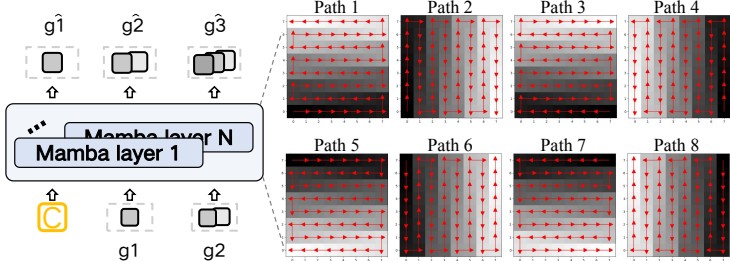

Figure 11: Sptial scan paths for Mamba.

## D Qualitative results with various resolutions.

Fig. 13 shows some generated samples with {256, 384, 512} resolutions. FlexVAR uses up to 256×256 resolution images for training, it can generate images with higher resolutions such as 384 and 512. The generated images demonstrate strong semantic consistency across multiple scales, and the higher resolutions display more detailed clarity.

## E Qualitative results with different VQVAE tokenizers.

**Image reconstruction.** We compare more image reconstruction results in Fig. 14. First, we encode the image into the latent space and performe multi-scale downsampling, then reconstruct the original image through the VQVAE decoder. It is evident that only our scalable VQVAE can perform image reconstruction at various scales.

**Generate images with GT prediction.** We visualize the generated samples with VQVAE tokenizers from VAR, Llamagen, and ours, corresponding to the $2^{nd}$, $3^{rd}$ and $5^{th}$ results in Tab. 6 in the main paper. As shown in Fig. 15, the VAR tokenizer, trained with a residual paradigm, fails to generate images under GT prediction; the generation samples of Llamagen's tokenizer are not up to the mark, due to its discrete tokens at intermediate steps being suboptimal.

## F Additional Visual Results.

We show more generated samples in Fig. 16.

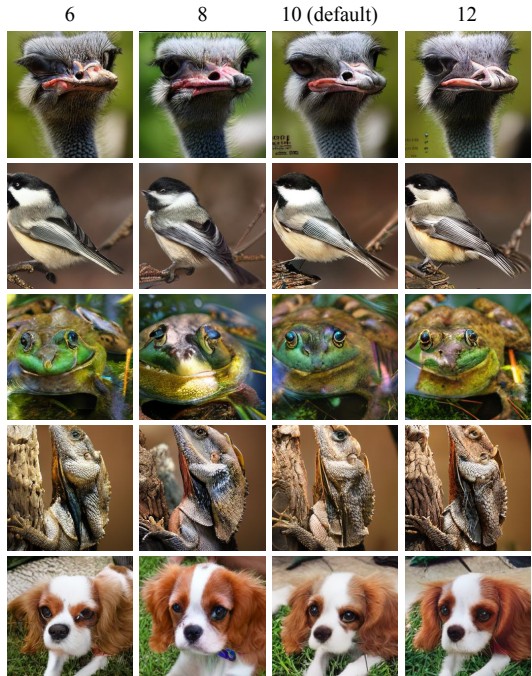

Figure 12: Some generated samples with {6, 8, 10, 12} steps. Note the model is trained with steps ≤ 10. More steps typically result in better image details. Zoom in for a better view.

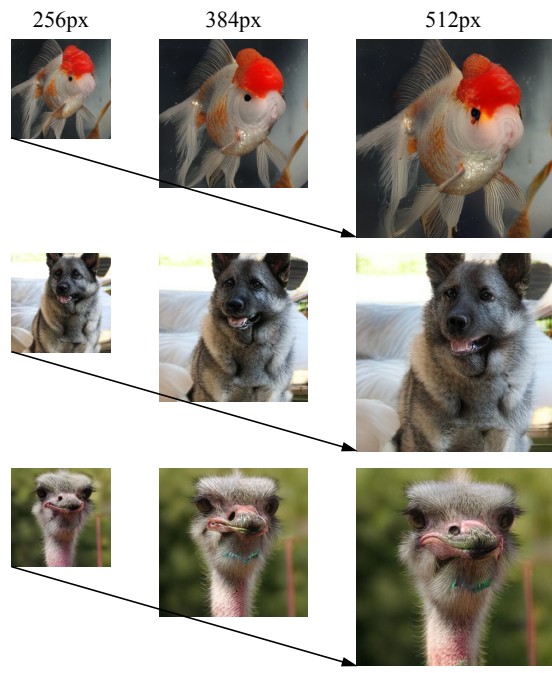

Figure 13: Some generated samples with {256, 384, 512} resolutions. Note the model is trained with a resolution of ≤ 256×256. Zoom in for a better view.

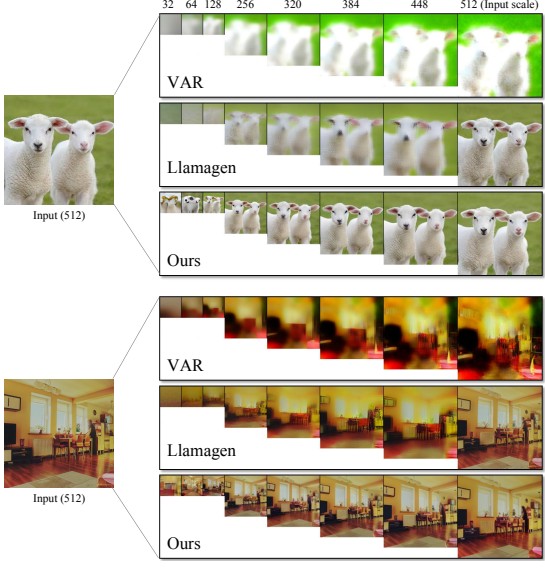

Figure 14: Compared with different VQVAE tokenizers [44, 41] for multi-scale reconstructing images, we downsample the latent features in VQVAE to multiple scales and then use the VQVAE Decoder to reconstruct images. We upsample images < 100 pixels using bilinear interpolation for a better view.

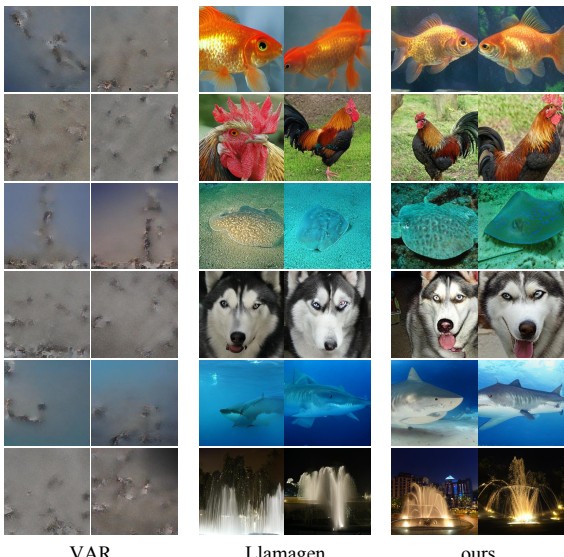

Figure 15: Some generated samples with different VQVAE tokenizers (Llamagen & VAR), corresponding to the $2^{nd}$ and $3^{rd}$ results in Tab. 6 in the main paper. We report the results with model scale -$d20$ trained 40 epochs ($\sim$ 70K iterations) on ImageNet-1K. Zoom in for a better view.

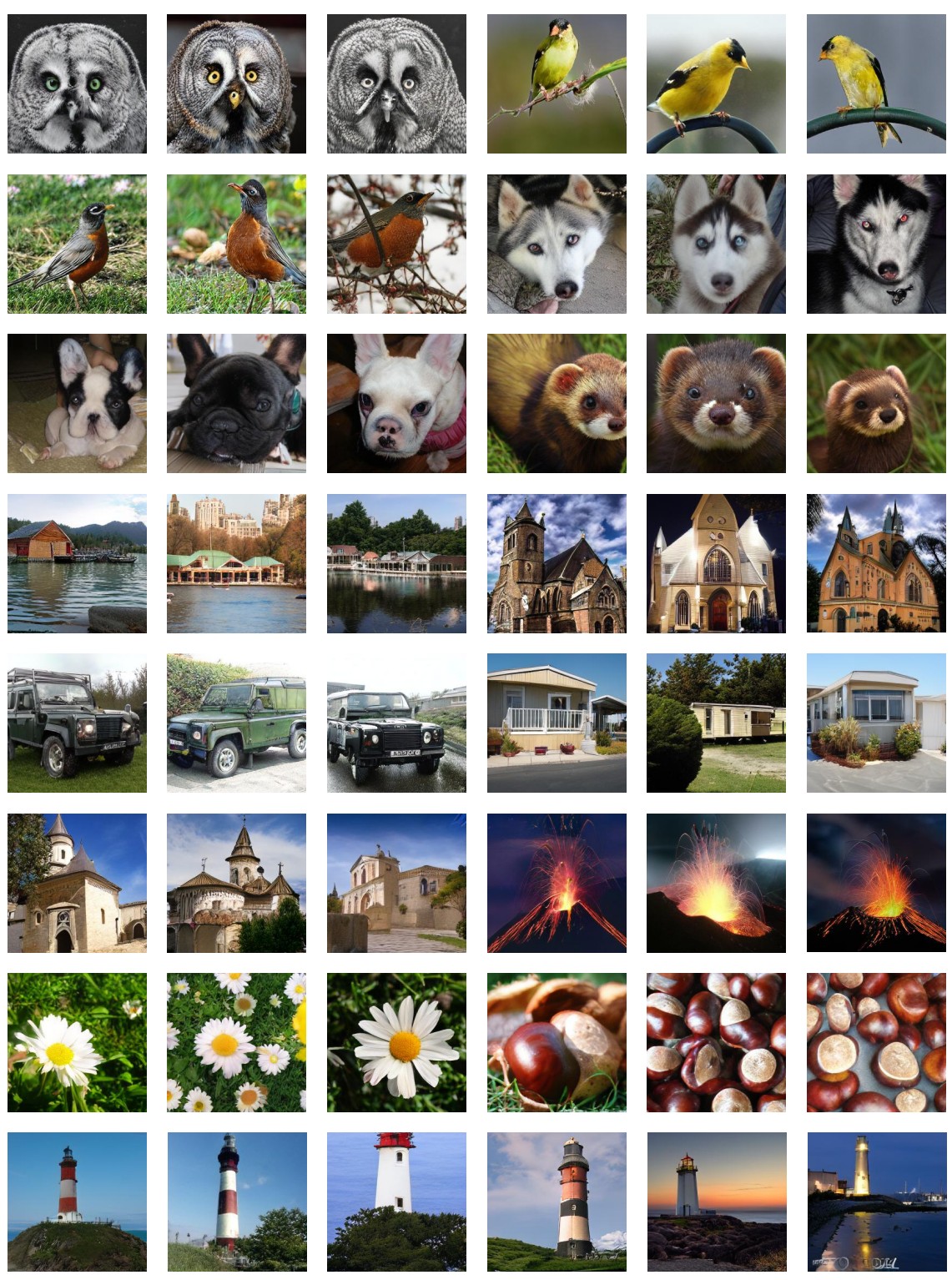

Figure 16: Some generated 256×256 samples.

