# OpenReview forum: "FlexVAR: Flexible Visual Autoregressive Modeling without Residual Prediction"
_NeurIPS.cc/2025/Conference — NeurIPS 2025 poster_

### Official Review · Reviewer_c8ru · 2025-07-02

**Clarity:** 3
**Significance:** 2
**Originality:** 2
**Rating:** 2
**Confidence:** 5

**Summary:**

FlexVAR overhauls multi-scale autoregressive image synthesis by skipping the usual residual-based updates and learning the actual latent codes at every resolution. It pairs a VQ-VAE tokenizer—built with multi-scale constraints so it can faithfully compress and reconstruct images at any size—with a Transformer that uses trainable 2D position embeddings to arrange those codes in sequence, all without being tied to a preset step order. The unified FlexVAR pipeline can then crank out semantically consistent images across a spectrum of resolutions and flexible generation schedules.

**Questions:**

1. Can you provide the performance of the densest scale {1,2,3,4,5,6,7,8,9,10,11,12,13,14,15,16} and sparsest scale like {1,4,16} or {1,2,4,8,16}? Will the results be upper bound and lower bound respectively?

2. Can your provide more analysis why you for adopting VAR’s scale configuration by default?

**Ethical Concerns:**

["NO or VERY MINOR ethics concerns only"]

**Final Justification:**

My ranking stays at rebuttal. My viewpoint  is not altered by  other reviewers or the rebuttal. FlexVar has no clear advantage over Var. Scaling is not clear.

**Limitations:**

Yes

**Quality:**

3

**Strengths And Weaknesses:**

Strengths:
1.	FlexVAR can produce images in arbitrary sizes and aspect ratios.
2.	FlexVAR achieves greater gains as additional scales are added.
Weakness:
1.	My main concern is the claim on lines 49–50—“The residual prediction relies on a rigid step design, restricting the flexibility to generate images with varying resolutions and aspect ratios.” In fact, VAR’s limitation comes from its fixed-length positional embeddings, not from residual prediction itself. The tokenizer of VAR with residual prediction can also reconstruct across arbitrary scale sets like {1, 2, 3, 5, 6, 7, 8, 9, 10, 12, 14, 16} (randomly picked).  FlexVAR’s added flexibility simply stems from removing those fixed-length 2D embeddings.
2.	FlexVAR offers no clear advantage over VAR. Although Table 2 shows marginal FID improvements at similar parameter counts, its IS scores are weaker, and Precision and Recall remain virtually unchanged. These results suggest that replacing residual prediction with direct ground-truth modeling does not yield meaningful performance gains, weakening the authors’ critique of the residual prediction paradigm.
3.	It seems FlexVAR’s image visual quality lags behind VAR’s—could you present side-by-side visual comparisons of FlexVAR and VAR outputs at the same generation step?
4.	FlexVAR’s scalability isn’t fully demonstrated: the authors only train up to 1 billion parameters, still trailing the top performance of VAR-d30.

---

> ### Author Rebuttal · Authors · 2025-07-29
>
> Thank you so much for acknowledging the strength of our method. We have carefully considered your constructive and insightful comments, and here are the answers to your concerns.
>
> **W1: Concern about the flexibility.**
> Residual modeling cannot achieve the flexibility of generating, as it relies on a weighted sum of all scales. This dependence requires a consistent scale design during both training and inference. Additionally, predicting residual does not allow for dropping scales because each scale is utilized during the weighted sum stage.
> We analyzed the differences between residual modeling and GT modeling in L50-L52: residuals at different scales often lack semantic continuity, and this implicit prediction approach may limit the model’s ability to represent diverse image variations.
>
>
> **W2: Replacing residual with ground-truth modeling does not yield meaningful performance gains**
> 1) FlexVAR demonstrates higher Inception Scores (IS) across different scales (d16, d20, d24) when compared to VAR. Our FlexVAR enhances IS further by utilizing zero-shot inference with more steps.
> 2) Precision and Recall metrics do not accurately reflect image quality; *i.e.*, the Precision score of VAR across different scales (VAR-d16, VAR-d20, VAR-d24) shows a decreasing trend: 0.84 -> 0.83 -> 0.82. Therefore, critiquing the ground-truth modeling paradigm based on IS, Precision, and Recall is not accurate.
>
>
>
> **W3: FlexVAR’s image visual quality lags behind VAR’s**
> Due to NeurIPS guidelines prohibiting anonymous links, we are unable to present side-by-side visual comparisons at this time. However, we have included a selection of generated samples in the *supplemental material* to showcase the visual quality of FlexVAR's images.
>
>
>
> **W4: FlexVAR’s scalability isn’t fully demonstrated**
> FlexVAR at scales d16, d20, and d24 outperforms the corresponding VAR models, which sufficiently demonstrates the scalability. The performance of FlexVAR-1B being weaker than VAR-d30 (2B parameters) should not be seen as our weakness. Due to limitations in computational resources, we are currently unable to train FlexVAR-d30. Therefore, to ensure a fair comparison, we only present models with parameters less than or equal to 1B, as explained in Line 198-199.
>
>
> **Q1: Performance of the densest scale and sparsest scale**
> We present the FID and IS of FlexVAR-d24 during inference with 3-steps ({1, 4, 16}), 5-steps ({1,2, 4, 8, 16}), 10, 13, 14, 15, and 16-steps in the table below. The results indicate that using too few inference steps (*e.g.*, 3 steps) results in poor performance, but there is a noticeable improvement in image quality with 6 steps. Both FID and IS achieve optimal values at step 14, followed by a slight decline in performance with more steps.
>
> |      | **3**  | **5**  | **6**  | **10** | **13** | **14** | **15** | **16** |
> |------|--------|--------|--------|--------|--------|--------|--------|--------|
> | FID  | 29.13  | 7.61   | 3.09   | 2.21   | 2.08   | 2.07   | 2.10   | 2.15   |
> | IS   | 75.65  | 159    | 260    | 299    | 315    | 316    | 316    | 312    |
>
>
> **Q2: Why adopt VAR’s scale configuration by default**
> We use the same scales as VAR for a fair comparison.

---

> > ### Comment · Reviewer_c8ru · 2025-08-05
> >
> > I am not convinced by the rebuttal for two reasons below. Hence I maintain my ranking of reject.
> >
> > 1.FlexVAR claim on lines 49–50—“The residual prediction relies on a rigid step design, restricting the flexibility to generate images with varying resolutions and aspect ratios.”       VAR's tokenizer has residuals VAR's generator has residuals and fixed-length absolution position embedding.  VAR's tokenizer is compatible with residuals, but the generator is not. Therefore, I believe the main limitation on flexible generation is the fixed-length absolute positional embedding, rather than the residuals.
> >
> >
> > 2. Since a lower FID indicates better image quality, FlexVAR-d24’s reported FID of 2.21 actually underperforms the original VAR-d24 result of 2.09. Besides, the FlexVAR paper report VAR-d24’s FID as 2.33—worse than the 2.09 reported in the VAR paper— the authors may have misstated or report lower VAR’s performance to exaggerate FlexVAR’s advantages on purpose.

---

> > > ### Author Response · Authors · 2025-08-07
> > >
> > > We respectfully disagree with the reviewer's comments that the main limitation on flexibility is the fixed-length positional embedding, and that we intentionally misreported the VAR results.
> > >
> > > 1) Even if the positional embedding is modified, image generation is still constrained by the residual weighted summation, and thus does not achieve flexibility.
> > > 2) Our reported FID results for VAR-d24 (2.33) are fully consistent with those reported by the VAR team on their official GitHub repository. Furthermore, concurrent work such as M-var [36] independently verifies similar VAR performance (see Table 2 in [36]), which further supports the accuracy of our reported numbers.

---

### Official Review · Reviewer_EER8 · 2025-07-03

**Clarity:** 3
**Significance:** 3
**Originality:** 3
**Rating:** 4
**Confidence:** 4

**Summary:**

This paper enhances the flexibility of VAR, which exhibits exceeding resolution generation, various image-to-image task support, and various autoregressive step adoption. The scalable tokenizer is robust to arbitrary resolution reconstructions and the flexVAR transformer models the ground-truth of each scale, which gets rid of the residual paradigm. The experiments show the effectiveness of the design.

**Questions:**

The design is simple yet interesting, providing new insights for efficiently generating high-resolution images. Check the weakness above.

**Ethical Concerns:**

["NO or VERY MINOR ethics concerns only"]

**Final Justification:**

After reading the rebuttal from other reviewers, I decide to maintain my original score since I think it provides a new insight in into how to generate a high-resolution image with only training on low-resolution ones.

**Limitations:**

yes.

**Paper Formatting Concerns:**

no.

**Quality:**

3

**Strengths And Weaknesses:**

**Strengths**
1) The simple yet effective design of modeling each scale instead of the residuals enhances the flexibility of VAR.
Equipped with multi-scale representation learning in VQ-VAE and scalable position embedding,  the visual autoregressive modeling on the GT images further enhances generation performance and speed.
2) The experimental results are sufficient and effectively convey the rationality of the design.

**Weakness**
1) Line 141 indicates the random scale sampling. Will it cause bias in specific resolution training. Will it cover high-resolution training.
2) When inference, how the attention mask is set. Will it be conditioned on all previous scale features. I suppose only the previous n-1 scale features are enough.
3) In Table 2. I am curious about why shifting the modelling paradigm from residuals to multi-scale modeling can bring benefits on gFID.
4) Line 170 indicates the training flops will be relatively high if sampling the resolution higher than 512.

---

> ### Author Rebuttal · Authors · 2025-07-29
>
> Thank you so much for acknowledging the strength of our method. We have carefully considered your constructive and insightful comments, and here are the answers to your concerns.
>
>
> **W1: Concerns about bias in specific resolution**
> The randomness inherent in the sampling process during training prevents the introduction of bias towards specific resolutions. Additionally, sampling only includes scales with resolutions $\leq$ 256. This approach does not encompass higher resolutions, thereby ensuring zero-shot inference at those higher resolutions.
>
> **W2: Attention mask during inference**
> During inference, each scale is conditioned on all previous scale tokens. We believe that the information from previous scales is valuable, as it helps prevent error accumulation.
>
>
>
> **W3: Why shifting the paradigm to GT modeling brings benefits to gFID.**
> We analyze the differences between two structures in L50-L52: residuals at different scales often lack semantic continuity, and this implicit prediction approach may limit the model’s ability to represent diverse image variations.
> Furthermore, the ablation study in Tab. 4 experimentally demonstrates that predicting ground truth is an effective approach.
>
>
> **W4: Concerns about the training flops.**
> We perform sampling at resolutions $\leq$ 256, deliberately avoiding larger scales such as 512.  As a result, compared to VAR, **no additional computational overhead** is introduced. Specifically, while VAR employs 10 fixed steps for sampling, we utilize random sampling within a range of 6 to 10 steps. Consequently, the training FLOPs may vary within a certain range but will not surpass the computational demands of VAR.

---

> ### Comment · Reviewer_EER8 · 2025-08-04
>
> Thanks for the rebuttal. Most of my concerns have been addressed. After reading other reviews, I decide to maintain my original score. The reasons are listed as follows:
> 1. I still have a question why simply training on resolutions smaller than 256, the model is able to generate 512*512 images. I think this part should be discussed.
> 2. The quality of zero-shot inference on 512*512 is not promising. And generating larger resolution (above 512) images is the main focus point.
>
> Overall, the proposed flexible design provides a new insight into how to generate a high-resolution image with only training on low-resolution ones.

---

> ### Author Response · Authors · 2025-08-05
>
> Thank you for the valuable discussion and constructive suggestions.
> Autoregressive generation models in the NLP domain inherently possess a degree of zero-shot capability, allowing them to extend to longer sequences beyond those seen during training. However, the design of VAR, specifically its use of residual weighted summation, completely eliminates the model's ability to generalize in a zero-shot manner to larger scales. In our proposed FlexVAR, we address this limitation by removing both residual prediction and residual weighted summation. This modification overcomes the fixed-resolution constraint of VAR and enables image generation at resolutions beyond those encountered during training.
> On the other hand, while FlexVAR demonstrates some zero-shot capability for generating high-resolution images, this does not imply that it can outperform methods that are fully supervised and trained directly at the target resolution. In Tab. 3, our evaluation at 512px resolution is conducted in a zero-shot setting, whereas all comparison methods are explicitly trained on 512px images.

---

### Official Review · Reviewer_B8QN · 2025-07-03

**Clarity:** 3
**Significance:** 3
**Originality:** 3
**Rating:** 4
**Confidence:** 4

**Summary:**

This paper challenges the dominant residual-prediction paradigm in VAR modeling. The authors propose FlexVAR, a new framework that instead predicts the ground-truth image representation at each generative step. This seemingly simple change, enabled by a novel scalable VQVAE and scalable positional embeddings, unlocks remarkable flexibility. Trained only on low-resolution images (≤256px), FlexVAR can zero-shot generate images at higher resolutions and arbitrary aspect ratios, perform various image-to-image tasks like inpainting and refinement, and dynamically trade speed for quality by varying its inference steps, achieving state-of-the-art results on ImageNet.

**Questions:**

1. Regarding the GT-prediction paradigm and error propagation: How does the model behave when it makes a significant error at a coarse scale (e.g., step 2 or 3)? Does the model learn to "correct" or ignore this incorrect context in later steps, or does the error tend to compound, leading to a flawed final image? An analysis of failure cases related to error propagation would be very insightful.

2. Regarding the GT-prediction paradigm and error propagation: How does the model behave when it makes a significant error at a coarse scale (e.g., step 2 or 3)? Does the model learn to "correct" or ignore this incorrect context in later steps, or does the error tend to compound, leading to a flawed final image? An analysis of failure cases related to error propagation would be very insightful.

3. The ability to use more inference steps to improve quality is a key advantage. Figure 6 shows FID improving up to 13 steps. What happens at the high end? Is there a point of diminishing returns (e.g., at 20 or 30 steps), or does performance begin to degrade due to compounding minor errors over a long generative chain? Understanding the practical upper limit of this flexibility would be valuable.

**Ethical Concerns:**

["NO or VERY MINOR ethics concerns only"]

**Final Justification:**

After reading the rebuttal, almost all concerns have been resolved. Therefore, I keep my original positive score.

**Limitations:**

yes

**Quality:**

3

**Strengths And Weaknesses:**

**Strengths:**

1.  This work makes a compelling case against a core assumption (the necessity of residual prediction) in a popular and powerful class of generative models. By proposing and successfully implementing a direct GTprediction paradigm, the paper offers a new, simpler, and more intuitive foundation for visual autoregressive modeling. This conceptual shift has a high potential impact on the field.

2. The ability of a model trained exclusively on low-resolution (≤256px) images to zero-shot generate high-quality 512px images (Table 3), handle arbitrary aspect ratios (Figure 5), and perform a suite of image-to-image tasks without any fine-tuning (Figures 7-9) is a significant breakthrough and a testament to the power of the GT-prediction paradigm.

3.  FlexVAR not only introduces a more flexible paradigm but also achieves excellent quantitative results. The 1.0B model surpasses its direct VAR counterpart and, when using more inference steps (a unique capability of the method), achieves a 2.08 FID on ImageNet 256x256, outperforming strong diffusion (DiT-XL/2) and autoregressive (AiM) models (Table 2). This demonstrates the method's effectiveness without sacrificing quality.

**Weaknesses:**

1. The paper's core paradigm, ground-truth prediction, lacks an explicit error-correction mechanism. In the residual-based VAR, each step can be seen as correcting the upsampled version of the previous scales. In FlexVAR, if the model generates an imperfect latent map at an early, coarse scale (e.g., predicts the wrong object silhouette), this error is fed forward as conditioning information to all subsequent, finer scales. The paper does not analyze or discuss this potential for compounding errors, which could lead to instability or inconsistencies in the final output, especially for complex scenes.

2. The proposed solution to make the VQVAE robust to varying latent scales is to train it on reconstructions from multiple, randomly downsampled versions of the latent map. While empirically shown to work (Figure 3), this approach feels more like a data augmentation trick than a fundamental architectural innovation.

3. In Table 2, the "Time" column provides a relative inference time comparison, which is very helpful. However, the best FID score for FlexVAR-d24 (2.08) is achieved with 13 steps and a relative time of 0.6, while the baseline models it is compared against, such as AiM, have a much higher relative time of 12. This comparison, while favorable for FlexVAR, does not compare models under an iso-time budget. A more direct comparison would involve evaluating baseline models under a similarly constrained time budget to better contextualize FlexVAR's efficiency.

---

> ### Author Rebuttal · Authors · 2025-07-29
>
> Thank you so much for acknowledging the strength of our method. We have carefully considered your constructive and insightful comments, and here are the answers to your concerns.
>
>
> **W1 & Q1: Analysis of error-correction**
> VAR predicts residuals at each step and weights the sum of all step residuals.  These residuals are not equivalent to errors, thus there is **no theoretical basis for VAR explicitly correcting errors**. Both VAR and our FlexVAR can see all previous scale tokens, allowing them to **implicitly correct previous errors**. From this perspective, predicting ground truth and predicting residuals show no significant difference in error accumulation. Additionally, Tab. 2 demonstrates that predicting ground truth is an effective approach.
> To further analyze error propagation,  we progressively added Gaussian noise within 1% of the maximum response scale at the early steps (*e.g.*, step \{1\} or steps \{1,2,3\}) to simulate the error generation process during inference. The FID results under different noise patterns are shown in the table below. It is evident that our FlexVAR consistently produces better quality than VAR under varying noise patterns.
>
> |         | **{-}** | **{1,}** | **{1,2}** | **{1,2,3}** | **{2,3}** |
> |------|---------|----------|-----------|-------------|-----------|
> | VAR-d20     | 2.95    | 3.47     | 3.54      | 2.91        | 3.18      |
> | FlexVAR-d20 | 2.41    | 2.35     | 2.33      | 2.49        | 2.53      |
>
> Note: FID is used for evaluation, where a lower score indicates better performance.
>
>
>
> **W2: The proposed VAE lack innovation.**
> We do not emphasize VAE as a fundamental architectural innovation. Our VAE follows the architecture of Llamagen and VQGAN, and the training of VAE is aimed at fitting the architecture that removes residuals. Its training primarily relies on data augmentation methods rather than structural improvements.
>
>
>
> **W3: The relative inference time favorable for FlexVAR.**
> There may be some misunderstandings about the relative inference time. We would like to clarify it as follows.
> Relative time is a time evaluation metric used by VAR, which **provides an intuitive reflection of inference time**. This metric is not favorable for FlexVAR.  Relative time represents the multiple of inference time required in comparison to VAR-d30 (as shown in Tab. 2 caption). *i.e.*, on an A100 GPU, the inference time for VAR-d30 is 0.5s. Therefore, multiplying 0.5 seconds by the corresponding relative time metric yields the actual image generation time for each model.
>
>
> **Q2: Zero-shot inference with more steps.**
> A 256-resolution image corresponds to a VAE latent feature of 16 × 16, allowing for a maximum of 16 sampling steps, ranging from 1 to 16. We present the FID and IS of FlexVAR-d24 during inference at steps 14, 15, and 16 in the table below, along with results for 6-step, 10-step, and 13-step for comparison. The results indicate that both FID and IS achieve optimal values at step 14, followed by a slight decline in performance with more steps.
>
> |      | **6** | **10** | **13** | **14** | **15** | **16** |
> |------|-------|--------|--------|--------|--------|--------|
> | FID  | 3.09  | 2.21   | 2.08   | 2.07   | 2.10   | 2.15   |
> | IS   | 260   | 299    | 315    | 316    | 316    | 312    |

---

### Official Review · Reviewer_zR9e · 2025-07-03

**Clarity:** 3
**Significance:** 2
**Originality:** 2
**Rating:** 4
**Confidence:** 4

**Summary:**

The paper proposes FlexVAR for image generation which builds on the VAR (visual autoregressive modelling) framework. Similar to VAR, the discrete token sequences are formed by using VQ tokens of different image scales. Different from VAR, the proposed VQVAE directly predicts the ground-truth of the image instead of the residual between different image scales. Experimental results show FlexVAR achieves similar performance to VAR.

**Questions:**

1. Could the authors clarify over the claim of FlexVAR outperforms VAR? VAR paper reports 2.09 FID on ImageNet, not 2.33 as reported in Table 2. At equivalent inference budget, FlexVAR performs worse than VAR. The better result requires more inference compute. In addition, if same technique of step dropping is applied to VAR during training, VAR might achieve similar or even better performance?

2. Could you provide a quantitive comparison of VQVAE performance trained by predicting groundtruth to that trained by predicting residual? Maybe the improvement is solely due to the quality of the discrete tokens?

3. Does FlexVAR scale similarly to VAR? From the trend of FlexVAR 3.05->2.41->2.21 v.s. VAR 3.3->2.57->2.09, it seems that FlexVAR is not making good use of the additional model capacity? Could the authors provide some discussions?

4. Could you explain in more detail how the aspect ratio is controlled during the inference process?

**Ethical Concerns:**

["NO or VERY MINOR ethics concerns only"]

**Limitations:**

Yes

**Quality:**

3

**Strengths And Weaknesses:**

**Strengths**

The paper is clearly written and experimental results seem strong.

By using scalable positional encoding, the paper demonstrates the ability of the model to generate images of different aspect ratios.

It confirms that using ground-truth works out similarly to using residual, as least up to a certain scale.

**Weaknesses**

The motivation is not super clear. Is there a theoretical reason why predicting ground-truth is better or worse than predicting residuals? Is predicting ground-truth unlock additional capabilities? The flexibility of generating images of different aspect ratios comes from using scalable positional embedding. Adaptive to more steps comes from randomly dropping steps during training time. These techniques are equally applicable to VAR.

---

> ### Author Rebuttal · Authors · 2025-07-29
>
> Thank you so much for acknowledging the strength of our method. We have carefully considered your constructive and insightful comments, and here are the answers to your concerns.
>
>
> **W1: Motivation.**
> Residual modeling cannot achieve the flexibility of generating, as it relies on a weighted sum of all scales. This dependence requires a consistent scale design during both training and inference. Additionally, residual modeling mandates using every scale in the weighted sum, thus preventing step dropping during training.
> We analyze the differences between two structures in L50-L52: residuals at different scales often lack semantic continuity, and this implicit prediction approach may limit the model’s ability to represent diverse image variations.
> Furthermore, the ablation study in Tab. 4 experimentally demonstrates that predicting ground truth is an effective approach.
>
>
> **Q1: VAR performance clarification.**
> 1) We perform evaluations using the official VAR model, and the results presented in Tab. 2 align with the performance reported by the VAR team on GitHub. Additionally, concurrent work M-var [36] also verifies the performance of VAR (Tab. 2 in [36]). Therefore, at an equivalent inference budget, FlexVAR demonstrates superior performance compared to VAR.
> 2) Step dropping is to enable the model to adapt to varying steps, which does not result in significant performance gains, as discussed in **W1**.
>
>
> **Q2: VQVAE performance.**
> Due to limited training resources, we train FlexVAR-VAE with fewer epochs and smaller batch sizes. We observe its reconstruction quality slightly inferior to VAR-VAE. Thus the improvement is not due to the quality of the discrete tokens, using a more robust FlexVAR-VAE might further improve the quality of generated images.
>
> |      | **Epoch** | **Batch-Size** | **rFID** |
> |------|-----------|----------------|----------|
> | VAR-VAE  | 20        | 768            | 1.92     |
> | FlexVAR-VAE (ours) | 10        | 128            | 3.79     |
>
>
> **Q3: Scaling capacity**
>
> As shown in Tab. 2, the FID trends when scaling up FlexVAR and VAR are similar (VAR: 3.55 -> 2.95 -> 2.33). Additionally, according to the FID formula, the decrease in FID score is non-linear, indicating that it becomes increasingly challenging to achieve further reductions as the score lowers:
> $$
> \text{FID}(P, G) = \|\mu_P - \mu_G\|^2 + \text{Tr}(\Sigma_P + \Sigma_G - 2 \cdot (\Sigma_P \Sigma_G)^{1/2})
> $$
> Where $\mu$ and $\Sigma$ represent the mean and covariance matrix of two distributions, respectively, and $\text{Tr}(\cdot)$ denotes the trace of the matrix.
>
>
>
>
> **Q4: Explication of aspect ratio generation.**
> We control the height and width for each scale through approximate rounding. *e.g.*, to generate an $H × W$ image, the corresponding VAE latent feature size is $h × w$, where $h = \frac{H}{8}$, $w = \frac{W}{8}$. We use VAR's default 10-steps ($K = ${$ 1, 2, 3, 4, 5, 6, 8, 10, 13, 16 $}) to determine the size of each scale. Consequently, the H × W image corresponds to ten scales with sizes {$(\text{int}(h \times \frac{i}{16}), \text{int}(w \times \frac{i}{16}))$}$_{i \in K}$. We will add more detailed explanations in the final version.

---

> > ### Comment · Reviewer_zR9e · 2025-08-05
> >
> > Thanks for providing additional details for the paper! Could you please add the VQVAE performance comparision in the appendix as well? Please also add a note for VAR, as Table 2 refers to VAR paper for the performance which is not reproducible if I understand correctly.
> >
> > Most of my concerns are addressed by the authors and I would like to maintain my score. The main reason being: while predicting groundtruth, flexible positional encodings and multi-scale generation may work together more naturally, the fixed scale in VAR is mainly a desgin choice, not a foundamental difference between predicting groundtruth and residual (e.g. one could train a multi-scale VQVAE with residual prediction).

---

> > > ### Author Response · Authors · 2025-08-05
> > >
> > > Thank you for the valuable discussion and constructive suggestions. We will include the performance of VQVAE in the final version and add explanations regarding VAR performance in Table 2. We greatly appreciate your time and effort in reviewing our submission.

---

### Decision · Program_Chairs · 2025-09-17

**Decision:**

Accept (poster)

**Comment:**

This paper presents a compelling and practical advancement in visual autoregressive modeling with FlexVAR. The core contribution is demonstrating that a simple but important shift—from residual to direct ground-truth prediction—and changing the positional encoding enables significant flexibility for image generation without compromising performance. This is a valuable finding for practitioners in the field.

There has been some difficulty in quantifying the performance of the baseline VAR model because the baseline paper publication and their updated github repository report conflicting metrics (2.09 vs 2.33 FID). For the purpose of this review I am willing to side with the authors; considering 2.33 the more trustworthy baseline.

The paper's main thesis is that abandoning the residual prediction paradigm is the key to unlocking flexibility. However, as Reviewer c8ru convincingly argues, the flexibility gains may instead stem from the paper's use of scalable positional embeddings, a feature distinct from the prediction target. The authors' rebuttal did provide some clarity, however I think additional ablations disentangle these two factors more carefully.

In summary however FlexVAR is a well-executed study that delivers a clear, useful result. The method's enhanced flexibility is a solid contribution and the discussion of pros/cons of ground-truth prediction will benefit the community. I am happy to recommend this paper for acceptance and look forward to seeing the discussion it inspires.